# DRC^2^-Net: A Context-Aware and Geometry-Adaptive Network for Lightweight SAR Ship Detection

**DOI:** 10.3390/s25226837

**Published:** 2025-11-08

**Authors:** Abdelrahman Yehia, Naser El-Sheimy, Ashraf Helmy, Ibrahim Sh. Sanad, Mohamed Hanafy

**Affiliations:** 1Department of Electrical and Computer Engineering, Military Technical College, Cairo 11766, Egypt; abdelrahman_yehia@ieee.org (A.Y.); issanad@ece.ubc.ca (I.S.S.); mehanafy@mtu.edu (M.H.); 2Department of Geomatics Engineering, University of Calgary, Calgary, AB T2N 1N4, Canada; 3Data Reception, Analysis and Receiving Station Affairs Division, National Authority for Remote Sensing and Space Sciences (NARSS), Cairo 1564, Egypt; akhelmy@narss.sci.eg

**Keywords:** SAR, ship detection, CNNs, YOLOX-Tiny, receptive field, attention mechanism, deformable convnets

## Abstract

Synthetic Aperture Radar (SAR) ship detection remains challenging due to background clutter, target sparsity, and fragmented or partially occluded ships, particularly at small scales. To address these issues, we propose the Deformable Recurrent Criss-Cross Attention Network (${\mathrm{DRC}}^{2}$-Net), a lightweight and efficient detection framework built upon the YOLOX-Tiny architecture. The model incorporates two SAR-specific modules: a Recurrent Criss-Cross Attention (RCCA) module to enhance contextual awareness and reduce false positives and a Deformable Convolutional Networks v2 (DCNv2) module to capture geometric deformations and scale variations adaptively. These modules expand the Effective Receptive Field (ERF) and improve feature adaptability under complex conditions. DRC^2^-Net is trained on the SSDD and iVision-MRSSD datasets, encompassing highly diverse SAR imagery including inshore and offshore scenes, variable sea states, and complex coastal backgrounds. The model maintains a compact architecture with 5.05 M parameters, ensuring strong generalization and real-time applicability. On the SSDD dataset, it outperforms the YOLOX-Tiny baseline with AP@50 of 93.04% (+0.9%), ${\mathrm{AP}}_{s}$ of 91.15% (+1.31%), ${\mathrm{AP}}_{m}$ of 88.30% (+1.22%), and ${\mathrm{AP}}_{l}$ of 89.47% (+13.32%). On the more challenging iVision-MRSSD dataset, it further demonstrates improved scale-aware detection, achieving higher AP across small, medium, and large targets. These results confirm the effectiveness and robustness of ${\mathrm{DRC}}^{2}$-Net for multi-scale ship detection in complex SAR environments, consistently surpassing state-of-the-art detectors.

## 1. Introduction

Synthetic Aperture Radar (SAR) is a high-resolution active sensing technology capable of operating under all-weather, day-and-night conditions. By exploiting microwave backscatter rather than ambient light, SAR is inherently robust to atmospheric interference such as clouds, fog, and precipitation, making it highly effective for maritime surveillance and target detection in complex environments. Nevertheless, cluttered backgrounds, geometric distortions, and the sparse distribution of ship targets in SAR imagery pose significant challenges for conventional object detection frameworks, often resulting in reduced accuracy and limited generalization. These constraints underscore the need for lightweight, context-aware detection models that are specifically tailored to the unique spatial and statistical properties of SAR data. Effective frameworks must balance real-time efficiency for deployment in resource-constrained environments with robustness to small-scale or partially occluded targets and ambiguous backscatter signatures. Addressing these challenges is essential to enable accurate, persistent, and automated monitoring across maritime, environmental, and defense-related applications [1,2,3].

Traditional ship detection methods, such as the Constant False Alarm Rate (CFAR) algorithm [3], have been widely employed due to their adaptive thresholding capability in clutter-rich maritime environments. While CFAR is effective in controlled or relatively simple scenarios, its performance often degrades in practical SAR applications. The algorithm relies on manually defined features and expert-set parameters, which increase processing time and limit scalability. In complex maritime conditions—characterized by varying sea states, heterogeneous backgrounds, and low signal-to-clutter ratios—CFAR frequently suffers from reduced accuracy and weak generalization [4]. This limitation stems from its dependence on accurate clutter modeling and continuous threshold calibration, both of which must dynamically adapt to changing environments to reduce false alarms and missed detections. With the growing complexity of SAR data and the increasing demand for real-time, high-precision maritime surveillance, traditional approaches such as CFAR alone are insufficient [5]. To address these issues, several enhanced CFAR variants and hybrid detection frameworks have been proposed, as briefly discussed in [2,6].

Recent advances in deep learning have greatly advanced SAR ship detection, with convolutional neural networks (CNNs) [7,8] demonstrating strong ability to learn hierarchical representations directly from raw data. Two main categories of CNN-based object detection architectures are commonly employed. The first, known as two-stage detectors, follows a coarse-to-fine strategy: region proposals are generated initially, followed by classification and bounding-box regression in a second stage. Representative models include Faster R-CNN [9], Libra R-CNN [10], and Mask R-CNN [11]. These methods typically achieve high detection accuracy but incur significant computational cost, which limits their suitability for real-time applications. The second category, single-stage detectors, performs classification and localization jointly in a unified pipeline. Examples include the YOLO family [12], SSD [13], and FCOS [14]. Owing to their end-to-end training design, single-stage detectors generally offer superior speed and simplicity, albeit sometimes at the expense of slightly reduced accuracy compared with two-stage approaches.

In SAR ship detection, key challenges stem from scale variation, occlusion, and directional backscattering, which complicate feature extraction. Background clutter, including speckle noise and sea surface texture, often leads to false alarms, particularly in lightweight models. Although deeper CNNs theoretically provide larger receptive fields, only a limited central region [15] significantly influences prediction. The fixed and spatially rigid receptive fields of CNNs make it difficult to adapt to ships of varying scales and orientations, a problem further amplified in coastal, port, and inland scenes where object–background confusion is common. These limitations highlight the need for tunable, multi-scale, and context-aware detection mechanisms.

To address these issues, recent works have explored diverse strategies. Zhao et al. [16] proposed the Attention Receptive Pyramid Network (ARPN), integrating Receptive Fields Block (RFB) and CBAM [17] to enhance global–local dependencies and suppress clutter. Tang et al. [18] introduced deformable convolutions with BiFormer attention and Wise-IOU loss to improve adaptability in complex SAR scenes. Zhou et al. [19] developed MSSDNet, a lightweight YOLOv5s-based model with CSPMRes2 and an FC-FPN module for adaptive multi-scale fusion. Cui et al. [20] enhanced CenterNet with shuffle-group attention to strengthen semantic extraction and reduce coastal false alarms. More recently, Sun et al. [21] proposed BiFA-YOLO, which employs a bidirectional feature-aligned module for improved detection of rotated and small ships. Overall, these studies emphasize that effective SAR ship detection requires models capable of balancing local detail sensitivity with global contextual awareness, particularly in cluttered and multi-scale maritime environments.

SAR ship datasets contain a high proportion of small targets with limited appearance cues such as texture and contour, making them challenging to detect. Detection performance is often hindered by the scarcity of features extracted from small ships and the mismatch between their scale and the large receptive fields or anchor sizes of conventional detectors. As mainstream frameworks typically downsample images to obtain semantic-rich features, critical information for small targets may be lost, leading to frequent missed detections [3,4,22].

To address these issues, several lightweight attention-augmented approaches have been proposed. Hu et al. [23] introduced BANet, an anchor-free detector with balanced attention modules that enhance multi-scale and contextual feature learning. Zhou et al. [24] proposed a multi-attention model for large-scene SAR images, enhancing detection performance in complex background environments. Guo et al. [25] further extended CenterNet with multi-level refinement and fusion modules to strengthen small-ship detection and suppress clutter with minimal overhead.

Despite progress with compact detectors that cut redundancy and incorporate attention for scale adaptability, reliable SAR ship detection remains difficult. Lightweight models, in particular, struggle with clutter, noise, and scale variation due to limited context modeling and rigid receptive fields. These gaps motivate the design of specialized, domain-tailored frameworks. To address these limitations, this paper proposes DRC^2^-Net, a compact and context-aware enhancement of YOLOX-Tiny. The proposed framework integrates lightweight semantic reasoning and adaptive spatial modules to strengthen feature representation, improve geometric adaptability, and enhance detection robustness in complex maritime scenes, all while maintaining high efficiency.

Although recent lightweight SAR detectors have achieved progress in reducing model complexity and incorporating attention mechanisms, they still struggle to capture global context and adapt to geometric variations in complex maritime clutter. Conventional convolutional structures, limited by fixed receptive fields, often fail to model long-range dependencies, leading to false alarms or missed detections—especially for small or irregular ship targets. These limitations motivate the need for a compact yet adaptive architecture specifically designed for SAR ship detection. To this end, ${\mathrm{DRC}}^{2}$-Net integrates RCCA and DCNv2 in a domain-specific manner, with DCNv2 selectively inserted at three critical neck locations identified through ablation studies. This design enhances geometric adaptability and contextual reasoning while maintaining a lightweight 5.05 M-parameter structure optimized for real-time maritime applications. The key contributions of this work are summarized as follows:Enhanced Semantic Context Modeling: Long-range spatial dependencies are captured by integrating a recurrent attention mechanism after the SPPBottleneck in the backbone. This placement enables semantic reasoning over fragmented, elongated, or partially visible ship structures, improving robustness against weak or ambiguous contours in complex maritime scenes.Adaptive and Flexible Receptive Fields: A novel DeCSP module embeds deformable convolutions into the bottleneck paths of three CSP layers in the neck, enabling dynamic, content-aware sampling. This design adapts to irregular ship scales and shapes while recovering shallow and boundary information often overlooked by conventional FPN-based fusion.Lightweight and Generalizable Detection Framework: The proposed DRC^2^-Net extends YOLOX-Tiny with targeted architectural enhancements while maintaining its lightweight nature (∼5.05M parameters). Evaluations on SSDD and iVision-MRSSD demonstrate strong generalization across varying resolutions, target densities, and clutter conditions, ensuring real-time performance suitable for maritime surveillance and edge deployment.

The remainder of this paper is organized as follows. Section 2 introduces the YOLOX-Tiny baseline and reviews the theoretical foundations of recurrent attention and deformable convolution. Section 3 presents the proposed DRC^2^-Net architecture, emphasizing its attention-aware and geometry-adaptive modules. Section 4 describes the experimental setup, datasets, evaluation metrics, and ablation studies conducted on SSDD and iVision-MRSSD. Finally, Section 6 summarizes the main findings and discusses potential avenues for future research.

## 2. A Lightweight Backbone

Accurate ship detection in SAR imagery requires broad contextual reasoning to suppress false alarms caused by sea clutter, together with detailed semantic discrimination to reliably localize weak or fragmented targets. Due to the frequent presence of coarse-resolution ships and highly dynamic maritime environments, traditional detectors often struggle to achieve an optimal balance between precision and efficiency. Current research increasingly focuses on lightweight, anchor-free frameworks tailored to the unique properties of SAR data. Such designs combine adaptive spatial sampling with long-range dependency modeling, enabling real-time operation in resource-limited settings while maintaining strong detection reliability [3].

### 2.1. YOLOX-Tiny Architecture

As introduced in the original “YOLOX: Exceeding YOLO Series in 2021” paper [26], the YOLOX family comprises six progressively larger variants: Nano, Tiny, S, M, L, and X, each balancing speed and accuracy to suit different deployment needs. In this work, we adopt YOLOX-Tiny as the baseline architecture due to its compact design and favorable trade-off between inference speed and detection accuracy. YOLOX adopts a center-based, anchor-free detection paradigm that localizes objects directly using key points, eliminating the reliance on predefined anchor boxes. This approach simplifies the detection pipeline, reduces computational complexity, and avoids the burden of extensive hyperparameter tuning [27].

The YOLOX network structure is composed of four main components: the input layer, the backbone for feature extraction, the neck for multi-scale feature fusion, and the prediction head. The overview of the YOLOX-Tiny model is illustrated in Figure 1. YOLOX framework uses CSP-Darknet as the backbone, leveraging Cross Stage Partial Networks (CSPNet) [28] to enhance gradient flow and reduce computational redundancy, and Spatial Pyramid Pooling (SPP) [29] for multi-scale feature extraction. CSP-Darknet offers a robust balance between accuracy and efficiency, making it a preferred choice in modern object detection models. YOLOX-Tiny is well-suited for real-time applications on resource-constrained platforms. The neck utilizes a Path Aggregation Network (PANet) [30], which fuses features through both top-down and bottom-up pathways. The top-down flow, inspired by Feature Pyramid Networks (FPN) [31], enriches semantic information, while the bottom-up path enhances spatial detail and localization precision, resulting in stronger representations across scales. The decoupled head design in YOLOX separates the object detection process into two distinct branches: classification and regression. This structural decoupling allows the model to independently optimize feature extraction for identifying object categories and for precisely localizing their spatial positions and dimensions. By minimizing task interference and distributing computational focus, the decoupled approach enhances both detection accuracy and operational efficiency, particularly important in complex SAR environments where fine-grained semantic discrimination and precise localization are critical.

### 2.2. Attention Mechanisms in SAR Ship Detection: (RCCA)

Detecting maritime targets in SAR imagery—particularly small, low-contrast, or partially visible ships—remains difficult. Small targets offer limited features that are easily lost due to receptive field mismatches, where anchors or kernels are disproportionately large. Discrete anchor scales further conflict with the continuous variation in ship size and orientation, reducing recall for targets between intervals. Incomplete targets, affected by sensor limits or background clutter, provide fragmented features that hinder accurate detection and classification [3]. CNN architectures, while effective in extracting semantic abstractions through deep hierarchical layers, often suffer from spatial resolution loss due to successive down-sampling. Consequently, small ships—occupying only a few pixels in SAR images—may lose discriminative features in deeper stages, leading to missed detections and reduced fine-grained recognition accuracy [2]. To address these limitations, attention mechanisms have been widely adopted for adaptively enhancing spatial and channel-wise features [32]. Transformer-based self-attention provides strong contextual modeling but remains computationally prohibitive for real-time or edge scenarios. Lightweight alternatives such as SE, SimAM [33], and CBAM are more practical but show clear drawbacks for SAR imagery: SE neglects spatial cues, while SimAM and CBAM depend on static pooling that limits long-range context. More recent approaches like Monte Carlo Attention (MCA) [34] attempt to capture global dependencies via stochastic sampling, yet face instability in cluttered maritime backgrounds. These challenges underscore the need for efficient, spatially aware attention mechanisms tailored to SAR ship detection.

To enhance pixel-wise representational capacity, the Criss-Cross Attention (CCA) mechanism was developed to efficiently capture contextual information along horizontal and vertical directions. Unlike traditional self-attention mechanisms, which incur high computational cost, CCA selectively aggregates features across rows and columns, significantly reducing complexity [35].

As illustrated in Figure 2, given an input feature map $H\in R^{C\times W\times H}$, the attention module begins by applying two 1×1 convolutional layers to produce the query (*Q*) and key (*K*) feature maps, where $\left\{Q,K\right\}\in R^{C^{\prime }\times W\times H}$ and $C^{\prime }<C$ for dimensionality reduction. After generating the query and key maps *Q* and *K*, an attention map $A\in R^{\left(H+W-1)\times (W\times H\right)}$ is computed via an affinity operation. For each spatial location *u* in *Q*, a feature vector $Q_{u}\in R^{C^{\prime }}$ is extracted. Correspondingly, a set $\Omega_{u}\in R^{\left(H+W-1\right)\times C^{\prime }}$ is constructed by collecting feature vectors from *K* that lie along the same row and column as *u*. Each element $\Omega_{i,u}\in R^{C^{\prime }}$ in this set represents a context vector. The affinity score between $Q_{u}$ and $\Omega_{i,u}$ is then calculated as shown in Equation (Equation 1):(1)$$d_{i,u}=Q_{u}\cdot \Omega_{i,u}$$
where $d_{i,u}\in D$ represents the correlation score between the query feature $Q_{u}$ and the corresponding key feature $\Omega_{i,u}$, for i=1,…,H+W−1. The resulting correlation matrix $D\in R^{\left(H+W-1)\times (W\times H\right)}$ captures the attention strength between each spatial location and its horizontal and vertical context.

A SoftMax operation is then applied along the attention dimension of *D* to normalize the values and produce the final attention map *A*. To adapt the features, another 1×1 convolution is applied to the input feature map *H*, producing $V\in R^{C\times W\times H}$. For each spatial position *u*, a feature vector $V_{u}\in R^{C}$ is extracted, along with a contextual set $\Phi_{u}\in R^{(H+W-1)\times C}$ comprising vectors from the same row and column. The final contextual representation is computed using an aggregation operation that fuses this information with the original feature at *u*. The final contextual representation is computed as shown in Equation (Equation 2):(2)$$H_{u}^{\prime }=\sum_{i=0}^{H+W-1}A_{i,u}\cdot \Phi_{i,u}+H_{u}$$
where $H_{u}^{\prime }$ is a feature vector in $H^{\prime }\in R^{C\times W\times H}$ at position *u*, and $A_{i,u}$ is a scalar value at channel *i* and position *u* in the attention map *A*. The contextual information $\Phi_{i,u}$ is integrated with the local feature $H_{u}$ to enrich the pixel-wise representation.

This mechanism enables a broader spatial receptive field and selectively aggregates relevant context via the spatial attention map. As a result, the enhanced features become more semantically expressive and robust, which is particularly beneficial for pixel-level tasks such as semantic segmentation [36]. While the CCA module enables efficient capture of horizontal and vertical dependencies, its single-pass operation may be insufficient to fully model the complex spatial relationships often encountered in SAR ship detection, where targets may appear fragmented or rotated within cluttered scenes.

To overcome this limitation, RCCA extends CCA by introducing iterative refinement across *R* loops. In the first loop, the input feature map *H* yields an updated representation $H^{\prime }$, with the same shape. A second pass then reprocesses $H^{\prime }$ to generate $H^{\prime \prime }$, effectively integrating contextual information from all pixels. By sharing parameters across loops, RCCA enhances global semantic reasoning while maintaining a lightweight footprint.

As illustrated in Figure 3, setting R=2 allows the module to aggregate full-image contextual information from all pixels, resulting in dense, context-rich feature representations. Let *A* and $A^{\prime }$ denote the attention maps in loop 1 and loop 2, respectively. With the help of a propagation function *f*, we can describe the information flow between any position *u* in $H^{\prime \prime }$ and any position θ in *H*.

Information can directly flow from θ to *u* when θ lies along the Criss-Cross path of *u*. However, when $\theta =(\theta_{x},\theta_{y})$ is not in the Criss-Cross path of $u=(u_{x},u_{y})$, the propagation is indirect:In Loop 1, θ transmits information to two intermediate positions: $\left(u_{x},\theta_{y}\right)$ and $\left(\theta_{x},u_{y}\right)$ (light green points), both of which lie on the Criss-Cross path of *u*.In Loop 2, these intermediate positions then relay the information to $u=(u_{x},u_{y})$ (dark green point).

This two-step message-passing mechanism enables θ to influence *u* even if it does not lie directly in its Criss-Cross path. As a result, RCCA captures long-range spatial dependencies and semantic context more effectively across the image domain.

### 2.3. Deformable Convolution Networks

In SAR-based maritime surveillance, ships often exhibit irregular or elongated geometries and may appear fragmented within cluttered sea or coastal environments. These hard-to-detect samples, combined with sparse and misaligned target distributions, pose significant challenges for lightweight detectors. While CNNs can extract hierarchical features, their grid-aligned and spatially rigid receptive fields—along with the limited scope of the effective receptive field—restrict adaptability to such complex cases. This mismatch between fixed sampling locations and actual ship structures frequently reduces detection accuracy in challenging SAR scenarios [15].

Unlike standard convolutions with fixed sampling grids, deformable convolutions dynamically adjust sampling positions based on local features, enabling the receptive field to adapt to the geometry of SAR ship targets. This flexibility enhances the model’s ability to extract relevant information from distorted or obliquely shaped ships, significantly improving robustness in cluttered or ambiguous maritime conditions.

To overcome the limitations of fixed-grid sampling in conventional convolutional layers, Deformable Convolutional Networks (DCNs) introduce a learnable offset mechanism that dynamically adjusts the sampling positions based on local content, as shown in Figure 4. This enhances the model’s ability to align with the actual structure of ship targets, which often vary in shape, scale, and orientation.

In a standard convolution, the output feature at location $p_{0}$ is computed as shown in Equation (Equation 3):(3)$$y\left(p_{0}\right)=\sum_{p_{n}\in R}\omega \left(p_{n}\right)\cdot X\left(p_{0}+p_{n}\right)$$
where *X* denotes the input feature map, ω represents the learnable weights, and $p_{n}$ is the relative offset from $p_{0}$ in the convolutional grid R (e.g., for a 3×3 kernel, R={(−1,−1),...,(1,1)}).

In contrast, deformable convolution introduces learnable offsets $\Delta p_{n}$ to each sampling location, enabling the network to shift the receptive field adaptively based on the input content. As illustrated in Figure 5, these offsets are predicted via parallel convolutional layers and organized into a 2N-channel offset map, where *N* is the number of sampling locations. Due to the fractional nature of $\Delta p_{n}$, bilinear interpolation is applied to compute precise feature values at the deformed positions [38].

This adaptive sampling improves the representation of irregular ship shapes and orientations commonly seen in SAR imagery.

The deformable convolution operation is thus expressed as shown in Equation (Equation 4):(4)$$y\left(p_{0}\right)=\sum_{p_{n}\in R}\omega \left(p_{n}\right)\cdot X\left(p_{0}+p_{n}+\Delta p_{n}\right)$$

In deformable convolution, each sampling location is adjusted by a learnable offset Δp, allowing the receptive field to shift flexibly according to the input features. Since Δp often results in fractional coordinates, the corresponding feature values are obtained via bilinear interpolation. However, this interpolation may inadvertently sample from irrelevant or noisy regions, potentially degrading the quality of the extracted features. To mitigate this issue, DCNv2 [39] introduces a modulation scalar $\Delta m_{k}\in \left(0,1\right)$ at each sampling location. This scalar acts as an adaptive attention weight, suppressing uninformative or noisy spatial regions by assigning lower values to less relevant sampling points.

The deformable convolution with modulation is mathematically defined as shown in Equation (Equation 5):(5)$$y\left(p_{0}\right)=\sum_{p_{n}\in R}\omega \left(p_{n}\right)\cdot X\left(p_{0}+p_{n}+\Delta p_{n}\right)\cdot \Delta m_{k}$$
where

$p_{0}$ is the current location in the output feature map;R denotes the regular grid of the convolution kernel;$\omega (p_{n})$ is the weight for the *n*-th location in the kernel;$\Delta p_{n}$ is the learnable offset for position $p_{n}$;$\Delta m_{k}$ is the modulation scalar applied to the sampled value.

The modulation coefficient $\Delta m_{k}$ is an adaptive attention weight that suppresses irrelevant spatial regions by assigning lower values to uninformative sampling points.

## 3. Methodology

### 3.1. Overall Network Structure of DRC^2^-Net

${\mathrm{DRC}}^{2}$-Net is a lightweight, context-aware detector built on the YOLOX-Tiny framework, noted for its balance of speed, compactness, and accuracy in real-time applications. While YOLOX-Tiny serves as a solid baseline, it encounters limitations in SAR imagery, especially when detecting sparse or partially visible ships within challenging maritime environments characterized by speckle noise and background artifacts. To address these challenges, this work introduces dual-modular enhancements applied to both the backbone and neck. The backbone preserves the four-stage design (Dark2–Dark5), producing feature maps C2–C5, with C3–C5 used as P3–P5 for detection, forming a hierarchical progression from high-resolution to high-semantic features. To strengthen deep semantic reasoning, the RCCA module is placed after the SPP bottleneck in Dark5, capturing horizontal and vertical dependencies through iterative refinement. This improves discrimination of ship targets while preserving the network’s lightweight efficiency.

In the neck, a bidirectional feature fusion strategy is adopted to enhance multi-scale ship representation. The top-down path first propagates deep semantic features to generate the feature pyramids P3, P4, and P5, which capture coarse but highly informative contextual cues. To preserve the fine-grained spatial details essential for detecting small or partially occluded ships, a bottom-up enhancement path subsequently aggregates shallow features upward, producing the refined outputs N3, N4, and N5. To further improve geometric adaptability within this fusion process, DCNv2 are strategically embedded into key CSP blocks. Unlike standard convolution, DCNv2 introduces modulated learnable offsets. This mechanism predicts not only spatial adjustments to the sampling grid but also a modulation mask that weights the contribution of each sampled value. Consequently, the network dynamically adjusts its receptive field in both position and intensity based on the local geometry of the input features, leading to superior adaptation to the diverse and complex shapes of maritime targets.

As illustrated in Figure 6, the input SAR image is first processed by the backbone (Dark2–Dark5), which extracts hierarchical features represented as C2–C5. From these, multi-scale pyramids P3–P5 are constructed, capturing small, medium, and large-target information. These pyramids are then fed into the neck, where bidirectional fusion generates intermediate maps N3–N5, enriching feature interactions across scales. Finally, three decoupled detection heads operate on N3–N5 to predict classification scores and bounding-box (BBox) regression. This end-to-end pipeline—from backbone encoding, through pyramid feature generation, to neck fusion and multi-head prediction—illustrates how the proposed framework transforms raw SAR imagery into accurate and scale-aware ship detections. Strategic enhancements within the backbone and neck further strengthen semantic continuity and geometric adaptability while maintaining the lightweight nature of the design.

### 3.2. RCCA Integration for Sparse Maritime Contextual Enhancement

In CNNs, spatial resolution diminishes with depth, reducing the ERF and causing loss of fine-grained detail. This is especially problematic for hard-to-detect samples such as small or partially visible ships, where complex backscatter and multi-path reflections obscure object boundaries and increase false negatives. To mitigate this, ${\mathrm{DRC}}^{2}$-Net integrates the RCCA module at the deepest backbone stage. As shown in Figure 7, RCCA is placed immediately after the SPPBottleneck block in Dark5, where semantic abstraction is high but spatial precision is weakened. By iteratively aggregating horizontal and vertical context, RCCA expands the ERF and restores continuity across distant regions while preserving essential spatial cues. This placement enables the network to better distinguish fragmented or low-contrast ships from background clutter with minimal overhead.

RCCA enhances semantic continuity through a lightweight two-pass refinement (R=2), following the setting validated in the official CCNet paper. Compared with single-pass CCA (R=1), the second recurrence enables dense contextual aggregation across all spatial locations without adding parameters or incurring significant computational cost. This extended context is particularly beneficial in SAR ship detection, where elongated or low-contrast vessels may overlap with clutter. Accordingly, integrating RCCA at the deep semantic stage of ${\mathrm{DRC}}^{2}$-Net strengthens spatial reasoning in challenging maritime scenes.

### 3.3. DCNv2-Enhanced Neck: Adaptive Geometry Modeling in Multi-Scale Fusion

To enhance spatial adaptability in multi-scale feature fusion, DRC^2^-Net integrates DCNv2 into the CSP modules of the neck. In contrast to standard convolutions that utilize a fixed grid, DCNv2 introduces learnable offsets which dynamically adjust the sampling positions based on the input’s local geometry. This allows the network’s receptive field to adaptively align with the diverse shapes and orientations characteristic of maritime targets.

As illustrated in Figure 8, deformable convolutions are integrated at three critical points in the neck: C3–N3 and C3–N4 in the bottom-up path, and C3–P3 in the top-down path, where accurate multi-scale feature alignment is essential. This enhancement is implemented through a custom Deformable CSP (DeCSP) layer, which preserves the original CSP architecture’s split–transform–merge strategy. Specifically, the standard 3×3 convolutions within the bottleneck blocks are replaced with DCNv2 layers, forming a DCN-Bottleneck. By embedding these DCN-Bottlenecks within the CSP structure, the network gains a superior capacity to capture rotated and distorted ship features, all while maintaining a low computational overhead. Instead of altering entire residual branches or replacing all convolutions—which provided only marginal benefits in preliminary trials—we adopt a selective design: only the 3 × 3 convolution inside the bottleneck block is substituted with a DCNv2 layer, forming a modular DeCSP block.

This modularity ensures that the original CSP structure can be preserved or extended with minimal architectural disruption. The two bottom-up insertions enhance early semantic fusion by adapting receptive fields to local geometric variations, while the top-down insertion reinforces high-level refinement, capturing global shape consistency. Together, these placements complement each other by balancing low-level adaptability with high-level contextual reasoning. By combining this geometric flexibility with the efficiency of CSP-Darknet, the design strengthens multi-scale feature fusion and significantly improves robustness to hard samples in SAR imagery.

This dual strategy leverages semantic attention and spatial adaptability in a complementary manner, effectively addressing both contextual ambiguity and geometric deformation. Importantly, the enhancements preserve the original YOLOX-Tiny detection head, ensuring that DRC^2^-Net retains its real-time inference speed and compact size.

## 4. Experiments

### 4.1. Dataset Description

To evaluate the proposed method, we employ two publicly available datasets: SSDD [40] and iVision-MRSSD [41].

The SSDD dataset contains diverse maritime scenes, including ports, offshore waters, and open seas, featuring ship types ranging from small fishing vessels to large tankers and container ships. Each image is standardized to 512×512 pixels, with spatial resolutions between 3–10 m, and annotations follow the PASCAL VOC format. The dataset is divided into training and testing sets in an 8:2 ratio, ensuring fair and reproducible evaluation.

As illustrated in Table 1, the two datasets provide complementary benchmarks for evaluating SAR ship detection performance. While SSDD captures typical maritime scenes with limited polarization and resolution diversity, iVision-MRSSD offers a broader multi-sensor representation across varied spatial resolutions, radar bands, and polarization modes. This diversity covers a wide range of maritime environments—from open seas to densely cluttered coastal zones—and includes inshore and offshore scenes as well as negative samples (ship-free images) to improve background discrimination. Together, these datasets support robust model development and reliable cross-domain benchmarking under realistic maritime conditions.

### 4.2. Implementation Settings

Since the SSDD dataset contains a relatively limited number of images, we adopt a transfer learning strategy by initializing the network with weights pre-trained on large-scale datasets. This enables the model to acquire general visual representations, thereby facilitating faster convergence and improved performance on the SAR-specific ship detection task. The baseline architecture is YOLOX-Tiny, configured with a depth multiplier of 0.33 and a width multiplier of 0.375, resulting in a compact network with approximately 5.05 M parameters.

All experiments were carried out on a Linux platform using PyTorch 2.0 and CUDA versions 12.1, with an NVIDIA Tesla T4 GPU 16 GB (NVIDIA, Santa Clara, CA, California). The input resolution was fixed at 512×512 pixels. Training was performed for 96 epochs, organized into four cycles of 24 epochs each, with a 5-epoch warm-up. Early stopping was applied with a patience of 12 epochs to mitigate overfitting.

Optimization was conducted using AdamW with an initial learning rate of $1.25\times 10^{-4}$ (scaled by batch size) and a weight decay of 0.05. A cosine annealing schedule was employed for dynamic learning rate adjustment. Data augmentation strategies included Mosaic (1.0), MixUp (0.3), and horizontal flipping (0.5). Evaluation was carried out every two epochs with a confidence threshold of 0.5. To ensure reproducibility, all experiments used a fixed random seed (42) and four data loading workers.

### 4.3. Evaluation Indicators

To comprehensively assess the detection performance of the proposed model, we adopt the standard COCO evaluation metrics [42], including multi-scale Average Precision (AP) and Precision–Recall analyses tailored for SAR ship detection.

Precision (P) is defined as the ratio of correctly predicted positive samples—true positives (TP)—to all samples predicted as positive, including false positives (FP). It reflects the model’s ability to minimize false alarms and is particularly critical in reducing false detections under cluttered SAR backgrounds, as shown in Equation (Equation 6):(6)$$P=\frac{TP}{TP+FP}$$

Recall (R) measures the proportion of actual positive samples correctly identified by the model, as shown in Equation (Equation 7). It captures the model’s capacity to detect all relevant targets:(7)$$R=\frac{TP}{TP+FN}$$

Precision and recall together provide a nuanced view of detection quality, especially important in maritime SAR scenarios where objects may be sparse or embedded in noisy backgrounds. The F1 score, defined as the harmonic mean of *P* and *R*, offers a comprehensive measure of a model’s classification performance, as expressed in Equation (Equation 8):(8)$$F1=\frac{2\times (P\times R)}{P+R}$$

Average Precision (AP) measures the area under the precision–recall (PR) curve, evaluating the trade-off between precision and recall across confidence levels, as formulated in Equation (Equation 9):(9)$$AP=\int_{0}^{1}P\left(R\right)\;dR$$

In this study, we report $AP_{50}$ (computed at a fixed Intersection over Union (IoU) threshold of 0.5) and the more comprehensive AP, averaged over multiple IoU thresholds from 0.5 to 0.95 in 0.05 increments. Additionally, we report $AP_{s}$, $AP_{m}$, and $AP_{l}$, corresponding to the model’s performance on small, medium, and large ship targets, respectively.

Intersection over Union (IoU) is a standard metric used to evaluate the accuracy of object detection models by comparing predicted bounding boxes to ground truth, as shown in Equation (Equation 10):(10)$$IoU=\frac{Area(B_{pred}\cap B_{gt})}{Area(B_{pred}\cup B_{gt})}$$
where $B_{pred}$ is the predicted bounding box and $B_{gt}$ is the ground truth. A higher IoU indicates better localization accuracy.

Furthermore, computational efficiency is assessed using the number of parameters (Params) and the number of floating-point operations (FLOPs). The total parameter count is the sum across all layers. For a convolutional layer, it is given as shown in Equation (Equation 11):(11)$$Params=\left(k_{h}\cdot k_{w}\cdot C_{in}\right)\cdot C_{out}$$
where $k_{h}$ and $k_{w}$ denote the kernel height and width, $C_{in}$ is the number of input channels, and $C_{out}$ is the number of output channels.

## 5. Results and Discussion

This section presents a comprehensive evaluation of the proposed ${\mathrm{DRC}}^{2}$-Net through three complementary analyses. These include ablation experiments validating the contribution of each proposed module, benchmark comparisons with state-of-the-art lightweight detectors on the SSDD dataset, and scene-level assessments on the iVision-MRSSD dataset. Together, these evaluations demonstrate the model’s architectural effectiveness, generalization capability, and operational robustness for SAR ship detection.

To ensure principled integration of the proposed attention mechanism, an initial comparison was conducted with representative alternatives. Specifically, CBAM and SimAM were inserted at the same spatial position as RCCA for fair evaluation. This preliminary experiment confirmed that RCCA provides stronger contextual reasoning and higher stability under cluttered SAR conditions, supporting its adoption in the final network design.

A structured ablation study was then performed to quantify the contribution of each architectural enhancement. Across three experiments and eight configurations, the analysis isolated the effects of contextual attention and deformable convolutions within both the backbone and the neck. The final configuration integrated the most effective modules into the complete ${\mathrm{DRC}}^{2}$-Net, confirming their complementary roles in strengthening multi-scale feature representation and improving overall detection accuracy.

Table 2 reports the results of Experiment 1, which examines the effect of contextual attention mechanisms embedded within the backbone. Using the default YOLOX-Tiny (BB-0) as the baseline, several enhanced variants were evaluated by inserting attention modules immediately after the SPP bottleneck in the Dark5 stage. These configurations include BB-1 (CBAM), BB-2 (SimAM), BB-3 (CCA), and BB-4 (RCCA).

To systematically evaluate the impact of attention mechanisms within the YOLOX-Tiny backbone, five configurations were tested by inserting different modules after the SPP bottleneck in the Dark5 stage. The baseline (BB-0), without attention, provided a solid reference with mAP@50 of 90.89%, AP of 61.32%, and F1-score of 94.57%, establishing the baseline representation capability for SAR ship detection. Integrating CBAM (BB-1) yielded the highest precision (97.30%) but reduced AP to 59.71% (−1.61%), indicating improved confidence but limited adaptability in cluttered SAR scenes. SimAM (BB-2) achieved an AP of 60.02% and the highest ${\mathrm{AP}}_{m}$ (89.63%) but only marginal recall improvement (92.86%), suggesting limited generalization across scales. CCA (BB-3) introduced criss-cross feature interactions, producing balanced results (AP = 61.21%, recall = 92.12%) yet underperforming for small and large targets. In contrast, RCCA (BB-4) delivered the best overall results: AP = 61.92% (+0.60%), recall = 93.04% (+0.55%), and ${\mathrm{AP}}_{l}$ = 84.21% (+5.26%), while maintaining strong precision (96.58%) and competitive mAP@50 (91.09%). RCCA thus demonstrates superior contextual reasoning and scale adaptability with minimal computational cost. Originally validated on the COCO segmentation benchmark, it shows consistent advantages when adapted to SAR imagery, confirming its suitability for integration into the final ${\mathrm{DRC}}^{2}$-Net architecture.

As shown in Figure 9, the Precision–Recall curve confirms this advantage: the RCCA-enhanced backbone sustains higher precision across a broad recall range, yielding superior AP@50. This visual evidence reinforces the quantitative findings, validating RCCA’s role in strengthening spatial–semantic representation and supporting its integration into the final ${\mathrm{DRC}}^{2}$-Net structure.

Table 3 presents the results of Experiment 2, which evaluates the impact of deformable convolutional enhancements within the neck while reusing the same backbone variants from Experiment 1. Three neck configurations were explored: NK-0 denotes the original YOLOX neck; NK-1 integrates two DeCSP blocks into the bottom-up path at C3_N3 and C3_N4; and NK-2 extends this design by adding a third DeCSP block in the top-down path at C3_P3. This setup isolates the contribution of the neck, particularly the influence of DCNv2, on multi-scale feature fusion and spatial adaptability, while maintaining consistency in the backbone structure across all variants.

The baseline configuration (NK-0), which employs standard CSPLayers, establishes strong performance with the highest AP (61.32%) and ${\mathrm{AP}}_{s}$ (89.97%), confirming its suitability for small-target detection. Introducing two DeCSP blocks in the bottom-up path (NK-1) increases precision to 97.48% and raises ${\mathrm{AP}}_{l}$ to 84.21% (+5.26% over baseline), indicating that adaptive sampling improves localization for larger and irregular ship targets. However, the overall AP slightly decreases (60.53%), suggesting that the deformable design reduces sensitivity to small-scale targets.

Extending the architecture with a third DeCSP block in the top-down path (NK-2) further boosts ${\mathrm{AP}}_{m}$ to 90.43% (+3.20%) and ${\mathrm{AP}}_{l}$ to 89.47% (+10.52%), demonstrating enhanced multi-scale refinement and geometric adaptability. Nevertheless, this configuration results in reduced mAP@50 (−1.29%) and a decline in ${\mathrm{AP}}_{s}$ (−3.24%), confirming that excessive deformability can weaken fine-grained detection in cluttered SAR backgrounds. Overall, these results highlight that while deformable convolutions benefit mid-to-large targets, careful balancing is required to preserve small-target accuracy.

Table 4 presents the results of Experiment 3, which integrates the most effective components identified in the earlier studies—namely, the RCCA-augmented backbone (BB-4) and the 3-DeCSP neck configuration (NK-2)—into a unified architecture, referred to as ${\mathrm{DRC}}^{2}$-Net. While the neck-only experiments indicated that deformable convolutions primarily benefit mid-to-large targets at the expense of small-scale accuracy, their combination with RCCA effectively balances this trade-off. In the final design, the three DeCSP modules work in harmony with RCCA, enhancing multi-scale representation without sacrificing the lightweight nature of the YOLOX-Tiny foundation.

Experiment 3 validates the complementary synergy between global contextual attention and deformable convolutional sampling. The integrated design enhances both semantic representation and spatial adaptability, yielding consistent improvements across scales. Specifically, ${\mathrm{DRC}}^{2}$-Net achieves gains of +0.98% in mAP@50, +0.61% in overall AP, and +0.49% in F1-score over the baseline YOLOX-Tiny. The improvements are particularly evident for small-object detection (${\mathrm{AP}}_{s}$: +1.18%) and large-object detection (${\mathrm{AP}}_{l}$: +10.52%). Overall, ${\mathrm{DRC}}^{2}$-Net represents a focused architectural refinement of YOLOX-Tiny, in which RCCA strengthens long-range contextual reasoning while DeCSP modules adaptively refine multi-scale spatial features. These enhancements produce a lightweight yet powerful SAR ship detector that balances efficiency with robustness, making it suitable for real-time maritime surveillance in complex environments.

### 5.1. Comparative Evaluation with Lightweight and State-of-the-Art SAR Detectors on SSDD

To comprehensively assess the performance of the proposed ${\mathrm{DRC}}^{2}$-Net, we benchmarked it against a range of representative object detectors. These include mainstream YOLO variants such as YOLOv5 [43], YOLOv6 [44], YOLOv3 [45], YOLOv7-tiny [46], and YOLOv8n [47], as well as lightweight SAR-specific models including YOLO-Lite [48] and YOLOSAR-Lite [49].

As summarized in Table 5, ${\mathrm{DRC}}^{2}$-Net achieves the highest F1-score of 95.06%, outperforming all baseline detectors. It also attains the highest precision (96.77%) and a strong recall (93.41%), highlighting its ability to minimize false positives while maintaining sensitivity to true targets. These results demonstrate that the integration of contextual reasoning and geometric adaptability in ${\mathrm{DRC}}^{2}$-Net leads to superior SAR ship detection performance across diverse conditions.

As summarized, the proposed ${\mathrm{DRC}}^{2}$-Net achieves superior performance compared with both general-purpose and SAR-specific lightweight detectors. It attains the highest F1-score of 95.06%, reflecting an optimal balance between precision (96.77%) and recall (93.41%), achieved with only 5.05M parameters and 9.59 GFLOPs. This demonstrates that the model maintains high accuracy while remaining computationally efficient. Compared to mainstream detectors such as YOLOv5 and YOLOv8n, which achieve precision above 95% but do not report F1-scores, ${\mathrm{DRC}}^{2}$-Net offers a more complete and balanced performance profile. Although YOLOv7-tiny exhibits the highest recall (94.9%), its relatively lower precision (92.9%) and lack of F1-score reporting limit a fair comparative assessment. In contrast, ${\mathrm{DRC}}^{2}$-Net consistently outperforms domain-specific lightweight models. YOLO-Lite achieves an F1-score of 93.39% and YOLOSAR-Lite 91.75%, yet both fall short of ${\mathrm{DRC}}^{2}$-Net’s accuracy while maintaining similar or larger parameter counts. With its compact architecture (5.05M parameters) and moderate computational cost (9.59 GFLOPs), ${\mathrm{DRC}}^{2}$-Net achieves an effective balance between detection accuracy and efficiency, demonstrating its suitability for real-time SAR ship detection in resource-constrained environments. While the model achieves an indicative inference rate of approximately 52 FPS on an NVIDIA Tesla T4 GPU, this value is hardware-dependent and not a definitive measure of architectural efficiency. Therefore, FLOPs and parameter count remain the primary, hardware-independent indicators of computational complexity, confirming ${\mathrm{DRC}}^{2}$-Net’s lightweight design. These results collectively establish a strong foundation for broader validation on diverse and higher-resolution datasets such as iVision-MRSSD, discussed in the following section.

### 5.2. Quantitative Evaluation on the iVision-MRSSD Dataset

We further evaluated the proposed model on the recently introduced iVision-MRSSD dataset, a high-resolution SAR benchmark released in 2023. In contrast to SSDD, iVision-MRSSD presents greater challenges due to its wide range of ship scales, dense coastal clutter, and highly diverse spatial scenarios, making it an appropriate benchmark for testing robustness in realistic maritime surveillance applications. A notable limitation of this domain is that many existing SAR ship detection models are not publicly available or lack detailed implementation specifications, hindering reproducibility. To ensure a fair and meaningful comparison, we therefore adopt uniform experimental settings wherever feasible and report the best available metrics as documented in the respective original publications. As shown in Table 6, recent lightweight detectors such as YOLOv8n (58.1%), YOLOv11n (57.9%), and YOLOv5n (57.5%) achieve the highest overall Average Precision (AP) on the iVision-MRSSD dataset. These results indicate notable progress in overall detection capability; however, they do not fully capture robustness across different target scales.

A detailed scale-wise evaluation highlights the advantage of the proposed ${\mathrm{DRC}}^{2}$-Net, which achieves 71.56% ${\mathrm{AP}}_{s}$, 84.15% ${\mathrm{AP}}_{m}$, and 78.43% ${\mathrm{AP}}_{l}$. These results significantly surpass competing baselines, particularly in detecting small- and medium-sized ships that are often missed by other models due to resolution loss and heavy background clutter in SAR imagery. In comparison, YOLOv8n and YOLOv11n report strong overall AP, but their ${\mathrm{AP}}_{s}$ values (51.5% and 52.1%, respectively) reveal persistent limitations in small-object detection.

Although ${\mathrm{DRC}}^{2}$-Net attains a slightly lower overall AP than YOLOv8n on the iVision-MRSSD dataset, this difference arises from the dataset’s heterogeneity and the model’s conservative confidence threshold, which prioritize precision and reliability under cluttered maritime conditions. This reflects ${\mathrm{DRC}}^{2}$-Net’s design focus on scale-aware robustness rather than aggregate metric optimization, aligning with the practical demands of SAR-based detection. By combining RCCA for global contextual reasoning with DeCSP modules for adaptive receptive fields, the framework maintains consistent accuracy across ship scales while ensuring efficient and reliable operation in complex maritime environments.

To qualitatively assess detection performance, representative scenes from the SSDD dataset are illustrated in Figure 10. Columns (a–i) cover diverse maritime conditions, including open-sea scenarios, nearshore environments, and multi-scale ship distributions within cluttered backgrounds. These examples emphasize the inherent challenges of SAR-based ship detection and provide visual evidence of the improvements achieved by the proposed ${\mathrm{DRC}}^{2}$-Net.

In all qualitative figures presented in this paper, each column corresponds to a distinct SAR scene, while the three rows represent different visualization layers: the top row displays ground-truth annotations, the middle row shows predictions from the baseline YOLOX-Tiny model, and the bottom row illustrates results from the proposed ${\mathrm{DRC}}^{2}$-Net. To maintain visual consistency, a unified color scheme is used across all examples: green boxes indicate ground-truth targets, red boxes denote correct detections, yellow boxes represent false positives, purple boxes highlight missed targets, and blue circles mark critical errors.

False alarms (yellow boxes) occur most frequently in open-sea and offshore scenes (Figure 10a–d), where wakes and wave patterns often resemble ships and mislead conventional detectors. ${\mathrm{DRC}}^{2}$-Net effectively mitigates these errors through deformable convolutions, which adapt receptive fields to better differentiate ships from surrounding clutter. Missed detections (purple boxes) are primarily observed in Figure 10e,f,h, typically involving small or low-contrast vessels. Notably, across all illustrated cases, ${\mathrm{DRC}}^{2}$-Net missed only one target in Figure 10g, demonstrating the effectiveness of RCCA in leveraging contextual cues to recover ambiguous or fragmented ships. Overall, these findings confirm that ${\mathrm{DRC}}^{2}$-Net delivers higher reliability by reducing false positives while enhancing sensitivity to challenging ship instances.

To further validate the generalization capability of the proposed ${\mathrm{DRC}}^{2}$-Net, instance-level visual comparisons were conducted across three representative sets of SAR scenes from the iVision-MRSSD dataset. These samples encompass data from six distinct satellite sensors and are grouped into three major scenarios, each highlighting specific detection challenges.

Figure 11 shows Scenario A (a–e), covering shorelines, harbors, and congested maritime zones with dense vessel clusters and coastal infrastructure. These conditions often trigger false alarms and mislocalizations, particularly for small, low-resolution ships affected by scale variation and background interference. While the baseline YOLOX-Tiny frequently misses or misclassifies such targets, the proposed ${\mathrm{DRC}}^{2}$-Net achieves more precise localization, especially near image edges, demonstrating greater robustness in challenging coastal environments.

Figure 12 presents Scenario B (a–f), which depicts densely packed ships in far-offshore environments. These conditions are characterized by low signal-to-clutter ratios, heavy speckle noise, and ambiguous scattering patterns, all of which make target visibility and discrimination difficult. In such challenging scenes, missed detections frequently arise from faint radar returns and poorly defined object boundaries. Compared with the baseline, the proposed ${\mathrm{DRC}}^{2}$-Net demonstrates stronger resilience to these issues, achieving more reliable detection under severe offshore clutter.

Figure 13 presents the final group of test scenes (Scenario C), featuring severe speckle noise, clutter, and ambiguous scattering patterns typical of moderate-resolution SAR imagery and rough sea states. These challenging conditions often lead to false positives and missed detections in baseline models. By contrast, the proposed model demonstrates stronger robustness, accurately localizing vessels despite degraded image quality and complex backgrounds.

In contrast, the proposed ${\mathrm{DRC}}^{2}$-Net demonstrates enhanced robustness by combining deformable convolutions with contextual attention, effectively suppressing spurious responses and improving target discrimination. The qualitative results confirm ${\mathrm{DRC}}^{2}$-Net’s ability to localize small and multi-scale vessels even under adverse imaging conditions, highlighting its generalization capability across offshore, coastal, and noise-dominant scenarios in the iVision-MRSSD dataset.

A quantitative summary of detection performance on the iVision-MRSSD dataset, expressed in terms of correct detections, false alarms, and missed targets, is provided in Table 7. The results demonstrate the improved performance of the proposed model across all scenarios. In Scenario A, ${\mathrm{DRC}}^{2}$-Net achieves 94.4% detection accuracy (17/18) compared with 61.1% for YOLOX-Tiny. In Scenario B, ${\mathrm{DRC}}^{2}$-Net reaches 95.7% (88/92) versus 83.7% for the baseline, reducing missed detections from 13 to 2. In the most challenging Scenario C, it attains 91.7% (22/24) versus 58.3% for YOLOX-Tiny. These results highlight the robustness of ${\mathrm{DRC}}^{2}$-Net, particularly in cluttered and noise-dominant SAR environments, achieving up to a +33.4% gain in detection accuracy over the baseline.

## 6. Conclusions

This paper presented ${\mathrm{DRC}}^{2}$-Net, a lightweight and geometry-adaptive detection framework tailored for SAR ship detection. Built upon YOLOX-Tiny, the architecture integrates RCCA into the deep semantic stage of the backbone to enhance global contextual reasoning, and introduces DCNv2 modules within CSP-based fusion layers to improve geometric adaptability. This dual integration strengthens semantic continuity and spatial flexibility while maintaining real-time efficiency and a compact 5.05 M-parameter design.

On the SSDD dataset, ${\mathrm{DRC}}^{2}$-Net achieves clear performance gains over the baseline YOLOX-Tiny. The AP@50 increases by +0.9% (to 93.04%), ${\mathrm{AP}}_{s}$ by +1.31% (to 91.15%), ${\mathrm{AP}}_{m}$ by +1.22% (to 88.30%), and ${\mathrm{AP}}_{l}$ by +13.32% (to 89.47%). These improvements are obtained with only 5.05 M parameters and 9.59 GFLOPs, confirming the model’s efficiency and suitability for real-time applications. Consistent improvements are also observed on the more challenging iVision-MRSSD dataset, where the proposed model achieves detection accuracies of 94.4%, 95.7%, and 91.7% across Scenarios A, B, and C, respectively. These results surpass YOLOX-Tiny and demonstrate strong generalization across diverse maritime conditions. Qualitative visualizations further reinforce the model’s robustness under cluttered and low-contrast SAR environments.

Importantly, no single architecture can optimally address all SAR ship detection tasks. Effective frameworks must balance accuracy, computational efficiency, and adaptability to mission-specific requirements and environmental constraints. The proposed ${\mathrm{DRC}}^{2}$-Net establishes a favorable trade-off among these factors, achieving high accuracy with minimal parameters and moderate computational cost, thereby providing a practical and deployable solution for real-time SAR ship detection. Future research will focus on model pruning, quantization, and rotated bounding-box prediction to further enhance deployment efficiency and detection precision in complex maritime environments.

## Figures and Tables

**Figure 1 sensors-25-06837-f001:**
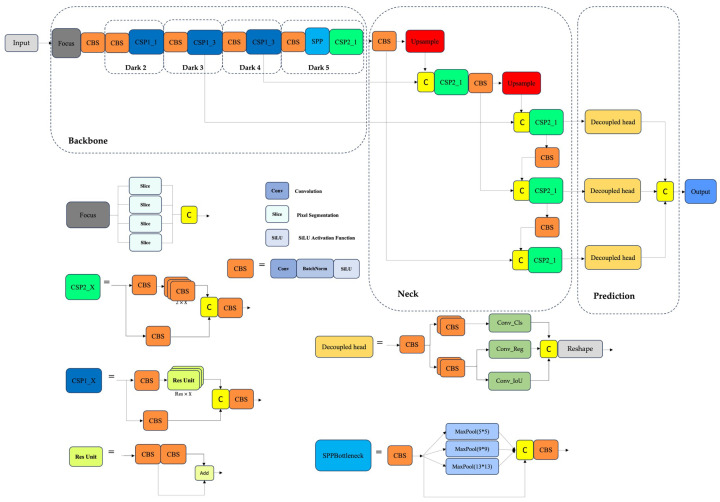
Overview of the YOLOX-Tiny model, showing its four main modules: input, backbone, neck, and detection head [26].

**Figure 2 sensors-25-06837-f002:**
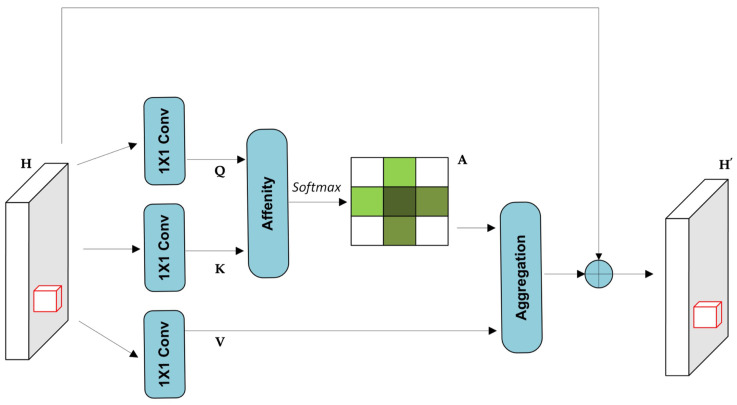
The Criss-Cross Attention module [35].

**Figure 3 sensors-25-06837-f003:**
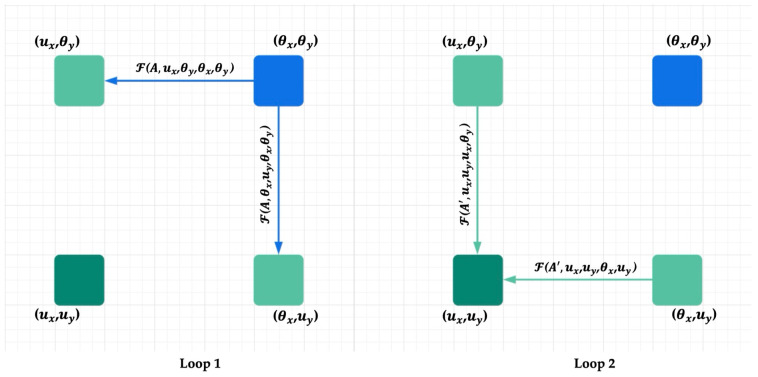
Information propagation when the loop number is 2 [35].

**Figure 4 sensors-25-06837-f004:**
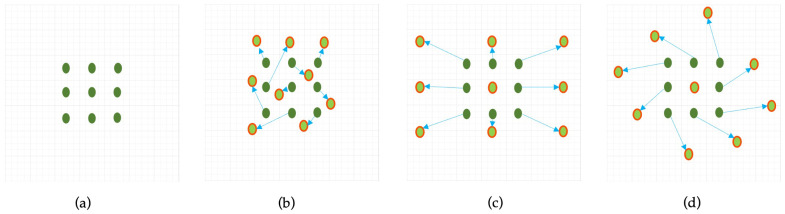
Illustration of sampling locations in 3 × 3 standard and deformable convolutions: (**a**) standard 3 × 3 convolution; (**b**) deformable convolution with learned offsets enabling adaptive kernel shapes; (**c**,**d**) specialized variants of deformable convolution [37].

**Figure 5 sensors-25-06837-f005:**
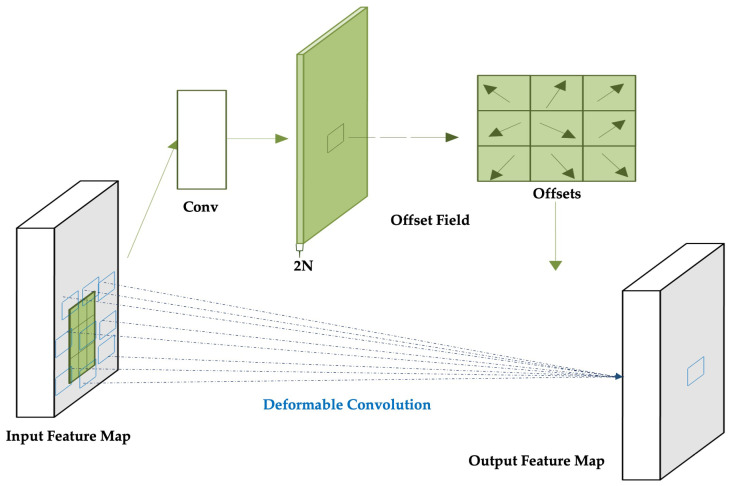
Illustration of a 3 × 3 deformable convolutional network. The offset field is derived from the input feature map and shares the same spatial resolution as the input [38].

**Figure 6 sensors-25-06837-f006:**
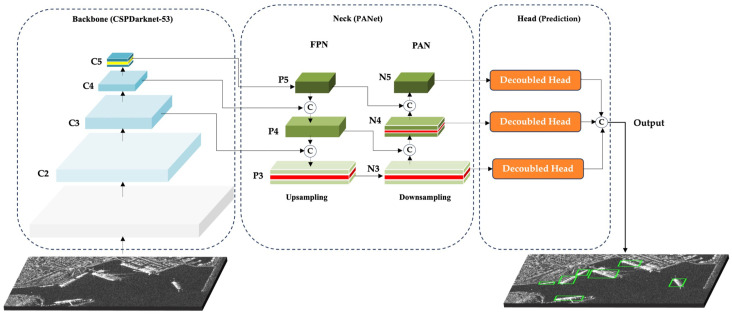
Overall architecture of the proposed ${\mathrm{DRC}}^{2}$-Net.

**Figure 7 sensors-25-06837-f007:**
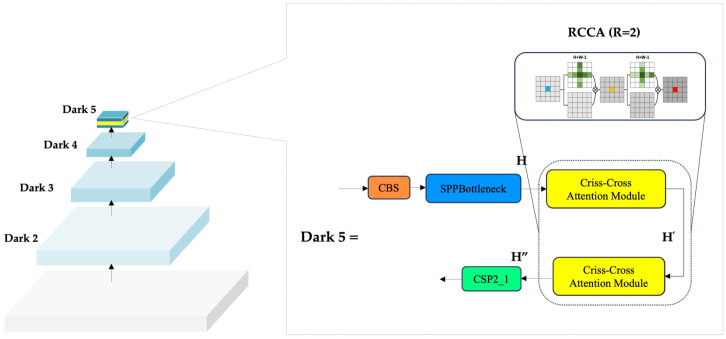
Structure of the proposed RCCA module integrated after the SPPBottleneck block.

**Figure 8 sensors-25-06837-f008:**
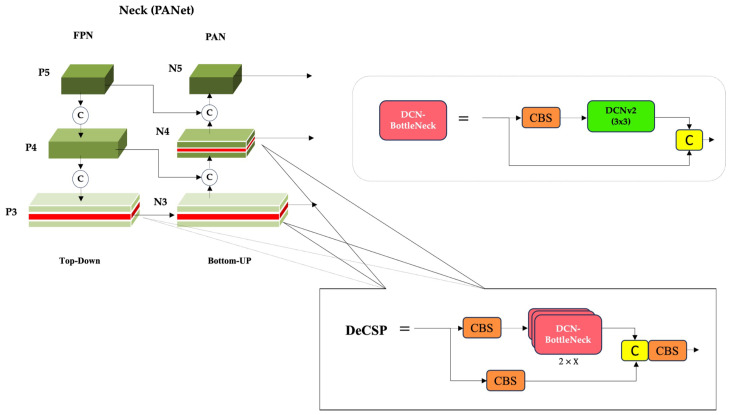
Structure of the proposed DeCSP modules integrated into the neck of ${\mathrm{DRC}}^{2}$-Net.

**Figure 9 sensors-25-06837-f009:**
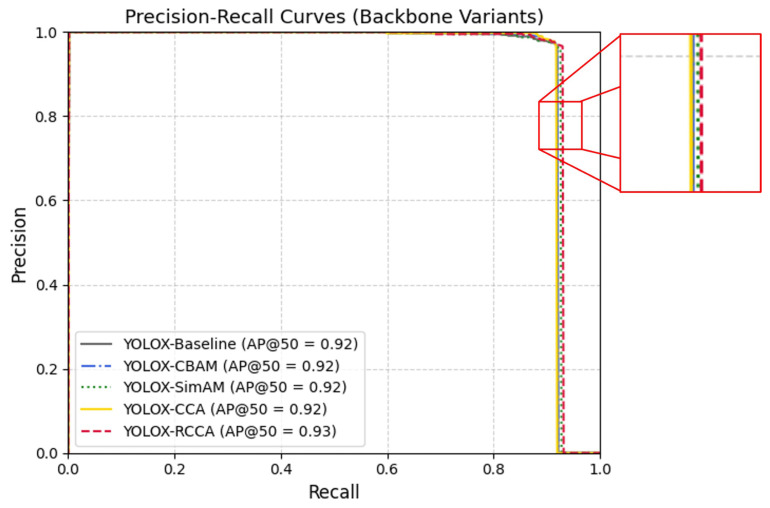
Precision–Recall (PR) curves comparing different attention-enhanced backbones.

**Figure 10 sensors-25-06837-f010:**
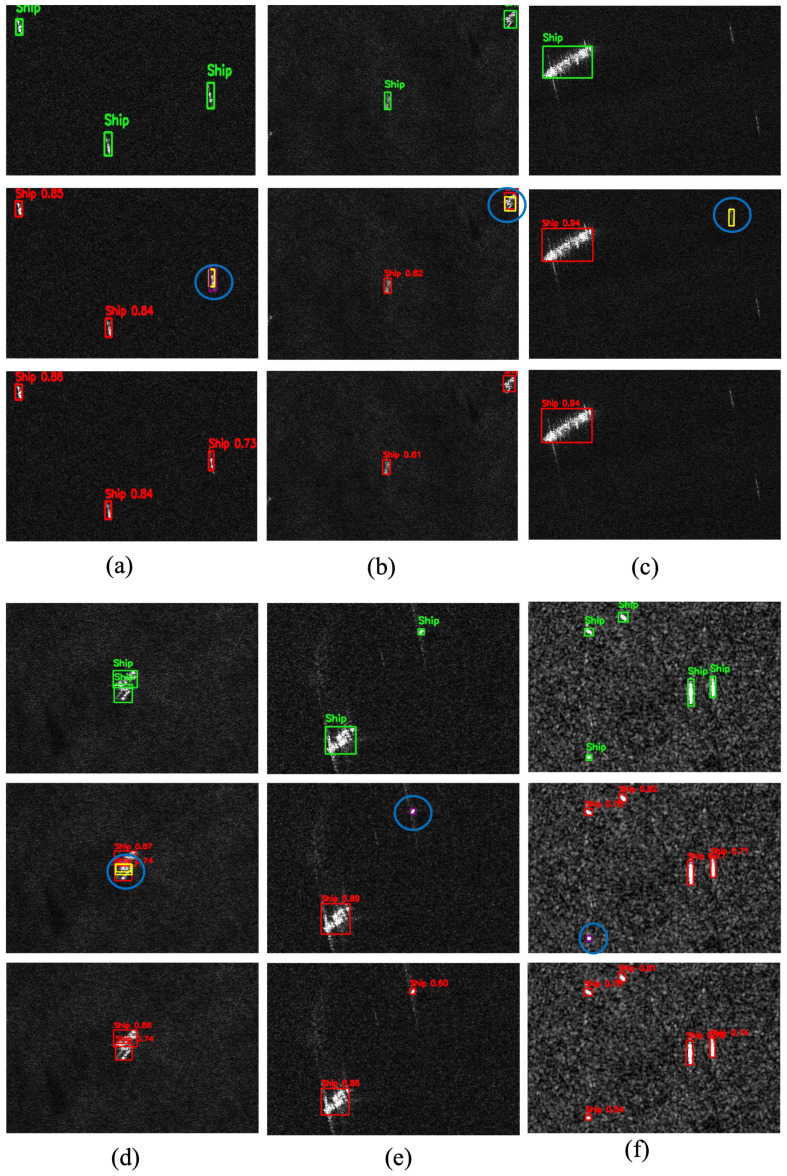
Visualization results on representative SSDD scenes. Groups (**a**–**c**), (**d**–**f**), and (**g**–**i**) correspond to diverse maritime environments, including open sea, nearshore waters, and multi-scale cluttered backgrounds.

**Figure 11 sensors-25-06837-f011:**
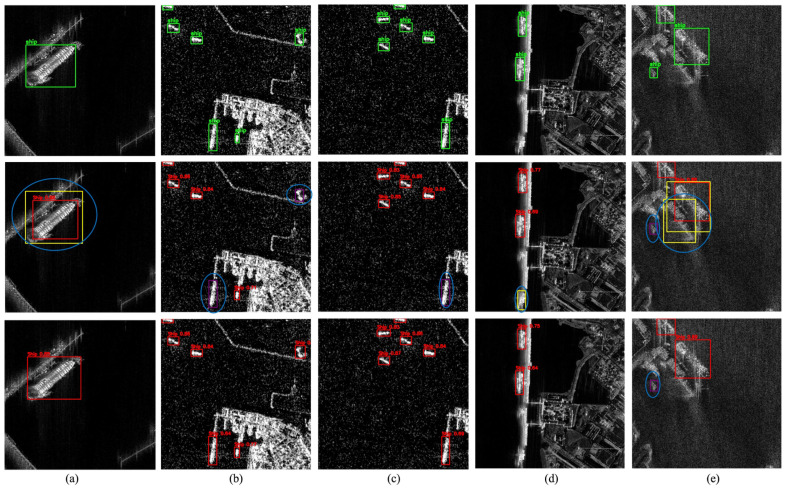
Visualization results for Scenario A (**a**–**e**) from the iVision-MRSSD dataset. Scenes represent shoreline and harbor environments with small-sized vessel clusters, occlusions, and coastal infrastructure.

**Figure 12 sensors-25-06837-f012:**
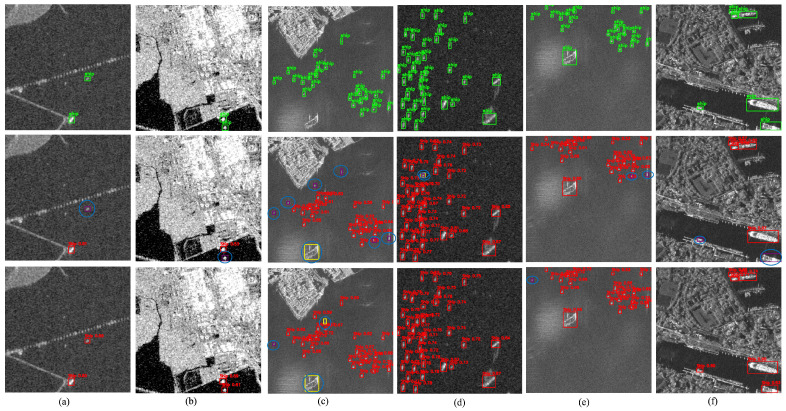
Visualization results of Scenario B (**a**–**f**) from the iVision-MRSSD dataset. The samples illustrate offshore clutter conditions, where densely distributed vessels and strong background interference increase the likelihood of false alarms and missed detections.

**Figure 13 sensors-25-06837-f013:**
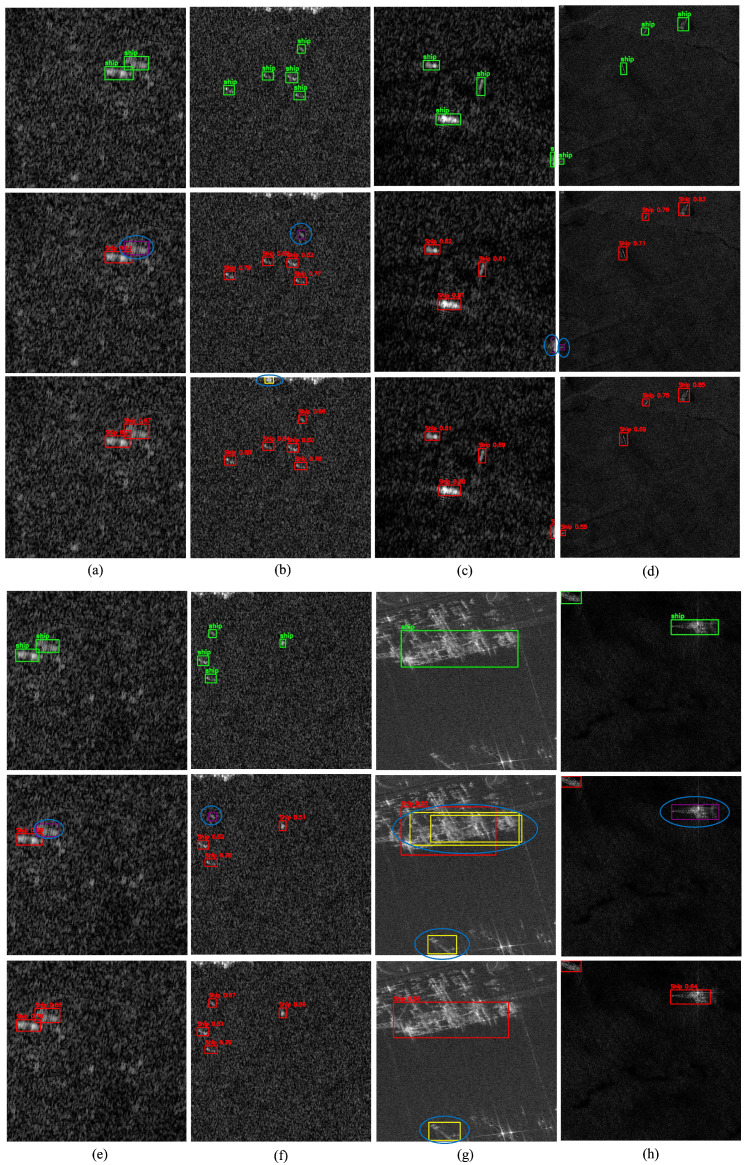
Visualization results of Scenario C (iVision-MRSSD) (**a**–**h**): scenes with severe speckle noise, textured clutter, and ambiguous scattering effects.

**Table 1 sensors-25-06837-t001:** Comparison of basic parameters between the SSDD and iVision-MRSSD datasets.

Attribute	SSDD	iVision-MRSSD
Data sources	RadarSat-2, TerraSAR-X, Sentinel-1	Capella Space, ICEYE, TerraSAR-X, Paz, ALOS-PALSAR, Sentinel
Polarization modes	HH, VV, VH, HV	Single, Dual, Quad
Bands	X and C	C, L, X
Resolution (m)	1–15	Multiple spatial resolutions
Category	Ship	Ship
Number of images	1160	11,590
Image size (pixels)	28×28–256×256	512×512
Number of ships	2456	27,885

**Table 2 sensors-25-06837-t002:** Results of the ablation studies of Experiment 1 on the backbone (best results in bold).

Backbone	mAP50	P	R	F1	AP	AP50	${\mathrm{AP}}_{s}$	${\mathrm{AP}}_{m}$	${\mathrm{AP}}_{l}$
BB-0 (Baseline)	90.89	96.74	92.49	94.57	61.32	92.12	89.97	87.23	78.95
BB-1 (CBAM)	90.58	**97.30**	92.94	94.84	59.71	92.12	89.09	88.30	84.20
BB-2 (SimAM)	90.46	96.20	92.86	94.50	60.02	92.76	89.97	**89.63**	84.20
BB-3 (CCA)	**91.59**	97.10	92.12	94.55	61.21	91.94	89.09	88.30	78.95
BB-4 (RCCA)	91.09	96.58	**93.04**	**94.78**	**61.92**	**93.04**	**89.97**	87.23	**84.21**

**Table 3 sensors-25-06837-t003:** Results of the ablation studies of Experiment 2 on the neck (best results shown in bold).

Neck	mAP50	P	R	F1	AP	AP50	${\mathrm{AP}}_{s}$	${\mathrm{AP}}_{m}$	${\mathrm{AP}}_{l}$
NK-0	90.89	96.74	**92.49**	94.57	**61.32**	**92.12**	**89.97**	87.23	78.95
NK-1	**90.97**	**97.48**	91.94	**94.63**	60.53	91.94	88.79	86.17	84.21
NK-2	89.60	97.09	91.58	94.25	61.20	91.58	86.73	**90.43**	**89.47**

**Table 4 sensors-25-06837-t004:** Results of Experiment 3 for the ${\mathrm{DRC}}^{2}$-Net (best results in bold).

Model	mAP50	P	R	F1	AP	AP50	${\mathrm{AP}}_{s}$	${\mathrm{AP}}_{m}$	${\mathrm{AP}}_{l}$
YOLOX-Tiny	90.89	96.74	92.49	94.57	61.32	92.12	89.97	87.23	78.95
**${\mathrm{DRC}}^{2}$-Net**	**91.87**	**96.77**	**93.41**	**95.06**	**61.50**	**93.04**	**91.15**	**88.30**	**89.47**

**Table 5 sensors-25-06837-t005:** Objective evaluation of recent lightweight detection models on the SSDD dataset (best results in bold).

Model	P (%)	R(%)	F1 (%)	Params (M)	FLOPs (G)
YOLOv5	92.11	89.84	90.96	9.12	24.04
YOLOv6	94.56	86.18	90.20	16.31	44.21
YOLOv3-tiny	94.00	90.40	92.22	12.00	18.90
YOLOv7-tiny	92.90	**94.90**	–	6.00	13.00
YOLOv8n	95.40	94.40	–	**3.00**	8.10
YOLO-Lite	96.28	90.63	93.39	7.64	–
YOLOSAR-Lite	92.30	91.20	91.75	**2.05**	**4.48**
**${\mathrm{DRC}}^{2}$-Net**	**96.77**	93.41	**95.06**	5.05	9.59

**Table 6 sensors-25-06837-t006:** Objective evaluation of recent detection models on the iVision-MRSSD dataset (best results shown in bold).

Model	AP (%)	${\mathrm{AP}}_{s}$ (%)	${\mathrm{AP}}_{m}$ (%)	${\mathrm{AP}}_{l}$ (%)
FCOS	43.5	39.3	50.1	39.6
ATSS [50]	53.2	46.7	61.4	59.2
YOLOv5n	57.5	51.1	66.4	69.5
YOLOv8n	58.1	51.5	66.7	**76.3**
YOLOv10n [51]	57.2	51.6	66.0	64.5
YOLOv11n [52]	57.9	52.1	66.1	69.6
**${\mathrm{DRC}}^{2}$-Net**	51.0	**71.56**	**84.15**	**78.43**

**Table 7 sensors-25-06837-t007:** Comparison of YOLOX-Tiny and ${\mathrm{DRC}}^{2}$-Net detection results on the iVision-MRSSD dataset across three SAR scenarios, showing the number and percentage of correct, wrong, and missed detections.

Scenario	Model	Correct (%)	Wrong	Missed	Accuracy (%)
A	YOLOX-Tiny	11	3	4	61.1
	${\mathrm{DRC}}^{2}$-Net	17	–	1	94.4
B	YOLOX-Tiny	77	2	13	83.7
	${\mathrm{DRC}}^{2}$-Net	88	2	2	95.7
C	YOLOX-Tiny	14	3	7	58.3
	${\mathrm{DRC}}^{2}$-Net	22	2	–	91.7

## Data Availability

The original contributions presented in this study are included in the article. Further inquiries can be directed to the corresponding author.

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
