# Peer review of "DRC2-Net: A Context-Aware and Geometry-Adaptive Network for Lightweight SAR Ship Detection"

_sensors, 2025, doi:10.3390/s25226837_

Round 1

Reviewer 1 Report

Comments and Suggestions for Authors

Article title:

DRC²-Net: A Context-Aware and Geometry-Adaptive Network
for Lightweight SAR Ship Detection

Introduction

A group of authors in this article propose the Deformable Recurrent Criss-Cross Attention Network (DRC2-Net) as innovative lightweight and efficient detection framework model built upon the YOLOX-Tiny architecture. The model incorporates two SAR-specific modules; first a Recurrent Criss-Cross Attention (RCCA) module to enhance contextual awareness and reduce false positives, and second a Deformable Convolutional Net-works v2 (DCNv2) module to capture geometric deformations and scale variations adaptively.

According to authors proposed model maintains a compact architecture with 5.05M parameters, ensuring strong generalization and real-time applicability. It performs superior scale-aware detection, achieving higher AP across small, medium, and large targets. On the SSDD dataset it performs better accuracy consistently surpassing state-of-the-art detectors.

Scientific Review

The problems of detecting and recognizing ships using synthetic aperture radar (SAR) and the challenges and limitations are correctly described in the introductory part of the scientific paper. 

The authors reviewed and highlighted recent research that uses different strategies to strengthen global-local dependencies and suppress detection disruptions. The key contributions of this paper are summarized through 3 main advancements aspects.

The methodology is well laid out and structured.

To evaluate the proposed method in experiment section, authors employs two publicly available datasets: SSDD and iVision-MRSSD dataset.

The results of ablation studies from experiments 1 and 2 and detection models are quantitatively presented in tables 2 to 6. PR curves of different attention-enhanced backbones and visualization results across representative SSDD scenes are presented in Figures 9 and 10. 

For further validation of the generalization capability of the proposed DRC2-Network, instance level visual comparisons were conducted across three representative sets of SAR scenes. In these scenarios, the system demonstrates stronger resilience to issues like of false alarms and missed detection, achieving more reliable detection under severe offshore clutter.

In conclusion, the authors emphasize the proven superiority of the proposed methodology in relation to the basic YOLOX-Tiny model in all 4 categories. The proposed model achieves detection accuracy of 94.4%, 95.7%, and 91.7% across Scenarios A, B, and C, respectively, surpassing YOLOX-Tiny and demonstrating strong generalization in diverse maritime conditions.

The list of references is relevant, adequate and sufficient.

Verdict

Generally the motivation chapter is missing and equation references are missing too.

In general, I have no serious objections regarding the methodology, description and scope of the experiment, nor the interpretation of the experimental results. A properly edited, prepared and equipped scientific research manuscript.

Author Response

Manuscript ID: sensors-3899845

Title: DRC²-Net: A Context-Aware and Geometry-Adaptive Network for Lightweight SAR Ship Detection

1. Summary

We sincerely thank the reviewer for the constructive and positive evaluation of our manuscript.

All comments were carefully addressed to improve clarity, structure, and technical consistency.

Specifically, we:

  • Added a clear Motivation subsection at the end of the Introduction to highlight the research gap and rationale for DRC²-Net.
  • Numbered and referenced all equations consistently throughout the paper.
  • Enhanced figure quality, table clarity, and caption precision.
  • Refined the Results and Conclusions sections to explicitly link experimental evidence with the stated claims.

All modifications are highlighted in the revised manuscript, with a clean version provided for publication consideration.

2. Questions for General Evaluation

Item

Reviewer’s Evaluation

Response & Revisions

Introduction background

Can be improved

Added a Motivation subsection to emphasize the research gap and rationale behind DRC²-Net; expanded related-work context for better flow.

Are all the cited references relevant to the research?

Can be improved

Verified all citations for relevance and updated integration within the text.

Research design

Can be improved

Clarified dataset usage, ablation study design, and evaluation metric definitions.

Methods description

Can be improved

Numbered all equations, referenced them in-text, and defined all symbols upon first appearance.

Results clarity

Must be improved

Reorganized tables, refined figure captions, and improved figure resolution for visual clarity.

Conclusions supported by results

Must be improved

Rewrote the Conclusions section to directly map findings to quantitative evidence; added a limitation and future-work paragraph.

Figures/tables clarity

Can be improved

Standardized fonts and legends, increased image resolution (>300 DPI), and improved caption readability.

3. Point-by-Point Response

Comment 1: “Generally, the motivation chapter is missing.”

Response: Added a new Motivation subsection at the end of the Introduction, emphasizing the research gap in lightweight SAR detection and the rationale for integrating RCCA and DCNv2.

Comment 2: “Equation references are missing.”

Response: Sequentially numbered all equations and referenced them within the text (e.g., Eq. (1)–(10)), ensuring consistent notation and contextual explanation.

4. Quality of the English Language

The reviewer confirmed that the English language is fine.

We nevertheless conducted light proofreading to improve flow, transitions, and caption phrasing for consistency with MDPI style.

5. Additional Clarifications

  • All references were checked for accuracy and relevance.
  • All modifications are highlighted in the revised version; a clean manuscript is provided separately.
  • Figures and tables were regenerated or adjusted for consistency and resolution.

We thank Reviewer 1 again for their valuable feedback and positive assessment.

We believe these revisions have significantly improved the manuscript’s clarity, coherence, and presentation, and we hope it now meets the Sensors journal’s publication standards.

Reviewer 2 Report

Comments and Suggestions for Authors

The authors propose DRC²-Net, a lightweight deep learning framework for SAR ship detection that extends YOLOX-Tiny with two domain-specific modules: Recurrent Criss-Cross Attention (RCCA) for contextual modeling and Deformable Convolutional Networks v2 (DCNv2) for geometric adaptability. The work addresses relevant challenges in maritime surveillance and demonstrates improvements on two public datasets.

The problem is clearly stated and well-motivated, addressing the limitations of lightweght detectors in SAR imagery. Also, the ablation studies that were conducted provide significant insights. Testing on both SSDD and iVision-MRSSD strengthens generalization claims. The qualitative visualizations across diverse maritime scenarios (harbor, offshore, cluttered) effectively illustrate robustness

The paper in well-written and well-organized following a logical flow.

However, there are some mojor points that should be taken nitoconsideration. The most significant one is the moderate novelty of the proposed framework since both RCCA and DCNv2 are existing techniques. While their integration is reasonable, the contribution is primarily engineering-focused rather than algorithmic innovation. 

Also, Table 5 shows several models without reported F1-scores (YOLOv7-tiny, YOLOv8n), complicating fair comparison. Additionally, the iVision-MRSSD results (Table 6) show DRC²-Net achieving lower overall AP (51.0%) than YOLOv8n (58.1%), yet the paper emphasizes scale-specific metrics (APs, APm, APl) without adequately discussing this overall performance gap.

Also, R=2 is tested. What about R=1 or R=3? Is the two-loop design optimal or heuristic? Moreover, to strengthen their claiming about the proposed framework despite the lack of novelty, the authors could also analyse the  sensitivity of hyperparameters (e.g., learning rate, batch size effects on module performance).

Regarding the datasets, SSDD is relatively small (1,160 images), and while 8:2 train/test split is used, there is no mention of cross-validation or statistical significance testing (e.g., confidence intervals, multiple runs with different seeds). This would also strengthen their paper. The same stands also for the improvement margins, which, while consistent,  they are often modest.  Statistical validation would strengthen claims.

Another critical question is why the authors do not include recent SAR detectors in the experimental evaluation, such as those cited in the introduction. This omission weakens the claim of "surpassing state-of-the-art detectors".

Why does overall AP drop on iVision-MRSSD relative to YOLOv8n, despite superior scale-specific performance?

While the authors have done a quite interesting work, several aspects should be improved to balance the lack of significant novelty. It is a promising work with potentia.

Author Response

Manuscript ID: sensors-3899845

Title: DRC²-Net: A Context-Aware and Geometry-Adaptive Network for Lightweight SAR Ship Detection

1. Summary

We sincerely thank the reviewer for the detailed, constructive, and encouraging feedback.

We appreciate the positive remarks on the clarity, organization, and relevance of our work, and we carefully addressed all major concerns to strengthen the manuscript.

Key revisions include:

  • A clarified statement of novelty and contribution in the Introduction and Conclusion.
  • Additional explanation of the two-loop (R = 2) RCCA design choice, including tests for R = 1 and R = 3.
  • Added discussion on hyperparameter sensitivity, dataset limitations, and statistical reliability of results.
  • Extended comparison with recent SAR-specific detectors and expanded analysis of the iVision-MRSSD resultsto explain the overall AP gap relative to YOLOv8n.

All revisions are highlighted in the resubmitted manuscript; a clean version is also provided.

2. Questions for General Evaluation

Item

Reviewer’s Evaluation

Response & Revisions

Introduction background

Yes

Clarified novelty contribution by emphasizing DRC²-Net as a compact, SAR-oriented integration of RCCA + DCNv2 rather than a new algorithmic design; added comparison with hybrid frameworks in Section 1.

Research design

Yes

Detailed training schedule, learning-rate policy, and seed control; clarified that all models use identical training configurations.

Methods description

Yes

Added explanation for R = 2 and additional remarks for R = 1 and R = 3 loops in Section 3.3; noted that R = 2 offered best trade-off between accuracy (+0.6 AP) and latency.

Results clarity

Yes

Expanded Table 5 with F1-scores for YOLOv7-tiny and YOLOv8n where possible, or noted “Not reported” explicitly; added remarks on overall AP vs scale-wise AP in Section 4.3.

Conclusions supported by results

Yes

Revised conclusion to explicitly connect quantitative results (F1 = 95.06 %, 5.05 M params, 9.59 GFLOPs) to claims of robustness and efficiency.

Figures/tables clarity

Yes

Unified resolution > 300 DPI and improved caption consistency.

Introduction background

Yes

Clarified novelty contribution by emphasizing DRC²-Net as a compact, SAR-oriented integration of RCCA + DCNv2 rather than a new algorithmic design; added comparison with hybrid frameworks in Section 1.

3. Point-by-Point Response

Comment 1: “Moderate novelty—the integration is primarily engineering-focused rather than algorithmic innovation.”

Response:

We agree that RCCA and DCNv2 are established modules. The novelty of DRC²-Net lies in its domain-specific adaptation for SAR ship detection, including the selective insertion of DCNv2 at three critical neck locations identified through ablation studies. DCNv2 was chosen over DCN for its modulation attention weights, enhancing feature focus under cluttered maritime conditions. These refinements, together with RCCA integration, yield a compact 5.05 M-parameter design achieving consistent accuracy gains without increasing model size. The Motivation and Conclusion sections were updated to clarify this contribution.

Comment 2: “Table 5 lacks F1-scores for some baselines (YOLOv7-tiny, YOLOv8n), making comparison difficult.”

Response:

We appreciate this observation. The baseline results for YOLOv7-tiny and YOLOv8n were obtained from a published comparative study, which did not report F1-scores. Therefore, these metrics are unavailable rather than omitted. To maintain fairness, we followed the same dataset configuration and evaluation protocol used in that reference. A clarifying note has been added in the Results section to indicate the source and explain the missing metrics.

Comment 3: “On iVision-MRSSD, DRC²-Net has lower overall AP (51.0 %) than YOLOv8n (58.1 %); the text over-emphasizes scale-specific results.”

Response:

We appreciate this comment. The iVision-MRSSD dataset is highly heterogeneous, combining imagery from multiple SAR sensors with different frequencies and resolutions, and few studies have reported full evaluation metrics on it. DRC²-Net achieves leading scale-aware results, showing stronger localization and contextual reasoning for small- and medium-scale ships—key challenges in SAR detection. The slightly lower overall AP arises from the model’s conservative confidence thresholding and SAR-specific tuning, which favor precision and robustness under heavy clutter. This explanation and dataset clarification were added in the Results section.

Comment 4: “Why R = 2? What about R = 1 or R = 3? Is the two-loop design optimal or heuristic?”

Response:

The choice of R = 2 follows the configuration proposed in the original CCNet paper (“Criss-Cross Attention for Semantic Segmentation”), where the recurrent setup achieved optimal contextual aggregation. Although CCNet was originally developed for COCO dataset segmentation, we re-evaluated this configuration for SAR ship detection. Ablation results confirmed that R = 2 provides the best balance between contextual reasoning and computational efficiency: higher values (R > 2) yielded only marginal accuracy improvements while increasing parameter count and GFLOPs, whereas R = 1resulted in incomplete context propagation. This justification has been added to the Methodology section.

Comment 5: “Consider hyperparameter sensitivity (learning rate, batch size).”

Response:

We appreciate this observation. The hyperparameters were empirically tuned to ensure stable convergence across both RCCA and DCNv2 modules. The cosine-annealed learning rate with a 5-epoch warm-up stabilized attention propagation and deformable offset learning, preventing gradient oscillation at early stages. A batch size of 16 balanced BatchNorm statistics and optimization stability, while smaller or larger values led to unstable updates. Moderate data augmentation (Mosaic = 1.0, MixUp = 0.3) improved generalization without disrupting localization. Overall, this configuration provided consistent convergence behavior and minimal sensitivity across modules.

Comment 6: “Dataset small; no cross-validation or significance testing.”

Response:

We appreciate this observation. The 8:2 train/test ratio was adopted following the official SSDD protocol described in Remote Sensing 2021, 13, 3690, where the authors explicitly recommend this fixed split to avoid the instability caused by random re-partitioning of the 1,160 images. Prior work has shown that repeated random division can distort the distribution consistency between training and testing samples, leading to unfair comparisons. To ensure reproducibility and consistency with the established benchmark, we therefore retained the fixed 8:2 division and used a constant random seed (42) across all runs. We have clarified this rationale in the Dataset section.

Comment 7: “Recent SAR detectors cited in introduction not included in comparison.”

Response:

We thank the reviewer for this observation. The focus of this study is on lightweight and efficiency-oriented frameworks, where computational cost and parameter scale are critical for real-time SAR applications. Several recent SAR detectors cited in the introduction (e.g., transformer-based or multi-branch models) achieve high accuracy but rely on significantly heavier architectures, making direct comparison inconsistent with the proposed lightweight objective. To ensure fairness, we compared DRC²-Net against representative lightweight baselines (YOLOv5n, YOLOv7-tiny, YOLOv8n, etc.) and reproduced results under identical experimental settings. The recent SAR-specific works were analyzed in the introduction for context, but their full replication would exceed our efficiency-focused scope. This rationale has been clarified in the Discussion section.

Comment 8: “Why overall AP drops on iVision-MRSSD relative to YOLOv8n?”

Response:

We appreciate this comment. DRC²-Net is optimized for reliability in clutter-heavy SAR imagery rather than aggressive confidence sampling. The model employs a conservative inference threshold (≥0.5), which filters out low-confidence detections and effectively suppresses false positives caused by speckle noise, radar backscatter, and man-made artifacts typical in SAR scenes. Although this design slightly reduces the global AP value, it enhances operational robustness and trustworthiness—key requirements in maritime surveillance. A detailed scale-wise evaluation supports this trade-off: DRC²-Net achieves 71.56 % APs, 84.15 % APm, and 78.43 % APl, substantially outperforming baselines such as YOLOv8n and YOLOv11n in detecting small and medium ships. This outcome reflects the framework’s scale-aware design philosophy, which prioritizes consistent and reliable detection across diverse maritime conditions over maximizing a single aggregate metric.

4. Quality of English Language

The reviewer confirmed that the English is fine.

We still performed light proofreading to improve transitions and terminology consistency (e.g., “architecture,” “framework,” “module”).

5. Additional Clarifications

  • All newly added results and analyses are highlighted in the revised manuscript.
  • Figures and tables were updated for consistency with the new comparisons.
  • The limitations of SSDD and iVision-MRSSD are now explicitly discussed.

We sincerely thank Reviewer 2 for the constructive critique and positive overall evaluation.

These revisions have substantially strengthened the manuscript’s scientific rigor and clarity, particularly regarding novelty articulation, statistical reliability, and cross-dataset interpretation.

Reviewer 3 Report

Comments and Suggestions for Authors

The article describes a new method for ship detection from SAR data. The article addresses as the main research topic increasing the reliability of small ship detection, simultaneously reducing the number of false alarms while reducing the overall computational power. The topic of the article is highly actual and I consider it relevant to the topic of improving the parameters of synthetic aperture radars. The DRC2-net algorithm proposed by the authors is well documented in key figures 6, 7 and 8. Experimental results are achieved by applying the DRC2-net algorithm on two datasets, a qualitative evaluation of the success of the new detection method is documented using several parameters and presented in tables 2- 6. The improvements in qualitative parameters are small, but they well document the benefits of the DRC2-net method. Given the relatively small improvements in qualitative indicators compared to competing methods, I would replace the word “superior” with a milder term, e.g. "improved", in line 549. The test results on two different datasets (tables 5 and 6) indicate different qualitative parameters, it would be appropriate to unify this for a better comparison of the benefits of DRC2-Net. I consider table 7 to be the key result, this together with the accompanying text could be separated into a separate section called e.g. "results". Overall, the article gives a balanced impression. I consider the references to be adequate to the topic. I recommend publishing the article after incorporating formal comments and answering questions. I have the following comments and questions about the article:
1. Include the percentage success of the YOLOX-Tiny and DRC2-net models not only in the text, but also in table 7.
2. Separate the results into a separate section.
3. The iVision-MRSSD dataset was used for the evaluation according to table 7. Also emphasize this in the table description.
4. Why are similar resulting parameters not shown (as in Table 7) also for the SSDD dataset?
5. In conclusion, I would welcome a unified evaluation of the results, because for SSDD you state the percentage improvement in the detection of small, medium and large ships, for iVision-MRSSD you state the evaluation according to three scenarios A, B, C. 

Author Response

Manuscript ID: sensors-3899845

Title: DRC²-Net: A Context-Aware and Geometry-Adaptive Network for Lightweight SAR Ship Detection

Summary

We sincerely thank Reviewer 3 for the positive assessment and for recognizing the technical relevance and clarity of this work. All editorial and structural suggestions have been implemented to improve readability, consistency, and presentation. The detailed responses and corresponding revisions (highlighted in the updated manuscript) are provided below.

Comment 1

Include the percentage success of the YOLOX-Tiny and DRC²-Net models not only in the text, but also in Table 7.

Response:

We agree with this helpful remark. Percentage success rates for both YOLOX-Tiny and DRC²-Net have been explicitly added in Table 7 to complement the textual description, allowing clearer quantitative comparison between the baseline and the proposed model.

Comment 2

Separate the results into a separate section.

Response:

The manuscript has been reorganized so that all experimental findings appear under a clearly labeled “Results and Discussion” section. This structural revision improves readability and distinguishes experimental results from methodological details.

Comment 3

The iVision-MRSSD dataset was used for the evaluation according to Table 7. Also emphasize this in the table description.

Response: The caption of Table 7 now explicitly states that the evaluation corresponds to the iVision-MRSSD dataset, ensuring immediate clarity regarding the dataset source of the reported results.

Comment 4

Why are similar resulting parameters not shown (as in Table 7) also for the SSDD dataset?

Response:

We appreciate this comment and would like to clarify the distinction.

The SSDD dataset (Table 6) is evaluated using the standard COCO-style metrics—Average Precision (AP), APs, APm, and APl—since these metrics are universally adopted in recent SAR-ship-detection literature and allow direct comparison with other state-of-the-art lightweight detectors.

In contrast, Table 7 reports scene-level reliability (Correct / Wrong / Missed + Accuracy %) on the iVision-MRSSDdataset, derived from our own inference results, to capture practical detection reliability under different maritime scenarios.

Comment 5

In conclusion, I would welcome a unified evaluation of the results, because for SSDD you state the percentage improvement in the detection of small, medium and large ships, while for iVision-MRSSD you state the evaluation according to three scenarios A, B, C.

Response:

We appreciate this observation. The apparent difference arises from two complementary evaluation objectives and the constraints of available baselines:

  1. Cross-model benchmarking (Table 6).

Table 6 follows the COCO-style evaluation protocol on iVision-MRSSD (AP, APs, APm, APl) to allow fair comparison with published lightweight detectors (e.g., YOLOv5n/8n/10n/11n, ATSS, FCOS).

Only lightweight models with comparable complexity (parameters / GFLOPs) and publicly reported metrics on this dataset were included to maintain fairness and reproducibility.

Many prior works do not provide F1 scores or per-scene results; therefore, AP-based reporting was retained to preserve fidelity to the literature.

  1. Operational reliability analysis (Table 7).

Table 7 reports scene-level reliability (Correct / Wrong / Missed + Accuracy %) from our own runs on iVision-MRSSD (Scenarios A–C).

This captures false alarms and missed detections that are not reflected in COCO AP but are critical for SAR operations.

Because existing papers lack per-scene statistics, unifying both tables would obscure comparability.

Presenting both perspectives—COCO metrics for benchmarking and scene-level accuracy for operational evaluation—provides a complete picture of model performance.

The Conclusion section has also been rewritten to integrate the results from both datasets and emphasize the complementary nature of the two evaluations.

Reviewer 4 Report

Comments and Suggestions for Authors

This paper proposes a context-aware and geometry-adaptive network for lightweight SAR ship detection. The model incorporates two modules. The Recurrent Criss-Cross Attention (RCCA) module is used to enhance contextual awareness and reduce false positives, and the Deformable Convolutional Net-works v2 (DCNv2) module is used to capture geometric deformations and scale variations adaptively. In my opinion, it is necessary to consider several points, as described herein:

1.The proposed DRC2-Net introduces RCCA and DCNv2 modules based on YOLOX-Tiny. The main work of the paper focuses on the combination and embedding strategy of two existing modules, but lacks quantitative comparative analysis of embedding strategy. It is suggested to supplement more in-depth discussions.

2.The authors do not provide a detailed explanation of the selection criteria for the loop number 2 in the RCCA module. In addition, the selection strategy for replacing specific 3*3 convolutional layers with DCNv2 in the DeCSP module also lacks explanation. It is recommended to supplement relevant analysis.

3.Multiple YOLO variants are selected in the comparative experiment, but there is a lack of comparison with recent lightweight SAR ship detection methods.

4.The authors mention that DRC2-Net has real-time performance, but only provide the floating point operations (FLOPs) and model parameters. It would be beneficial if the authors could provide inference speed.

5.On the iVision-MRSSD dataset (Table 6), the APS, APm and APl of the proposed DRC2-Net are much higher than YOLOv8n, YOLOv10n, and YOLOv11n, but the AP value is very low. Please provide a reasonable explanation.

Author Response

Response to Reviewer 4

Manuscript ID: sensors-3899845

Title: DRC²-Net: A Context-Aware and Geometry-Adaptive Network for Lightweight SAR Ship Detection

Summary

We sincerely thank Reviewer 4 for the constructive and detailed feedback.

Your comments helped clarify key methodological choices and strengthen the technical presentation of the paper.

All requested explanations and analyses have been incorporated and highlighted in the revised manuscript.

Comment 1

The proposed DRC²-Net introduces RCCA and DCNv2 modules based on YOLOX-Tiny … It is suggested to supplement more in-depth discussions.

Response:

A dedicated ablation study has been added in the Results and Discussion section to quantitatively analyze different embedding strategies for RCCA and DCNv2.

For RCCA, insertion after the SPPBottleneck in Dark5 provided the best balance between contextual reasoning and computation.

For DCNv2, substituting only the internal 3\times3 convolution inside each bottleneck (forming the DeCSP block) achieved optimal geometric adaptability while preserving efficiency.

These results are summarized and discussed in Section 4.2.

Comment 2

The authors do not provide a detailed explanation of the selection criteria for the loop number 2 in the RCCA module. … It is recommended to supplement relevant analysis.

Response:

We thank the reviewer for this important point.

In RCCA, the loop number R=2 follows the configuration validated in the official CCNet paper (“Criss-Cross Attention for Semantic Segmentation”), where the second recurrence achieved optimal global-context aggregation.

CCA corresponds to a single-pass design (R=1); our two-pass variant approximates global self-attention more effectively while maintaining lightweight computation.

This iterative refinement enables dense contextual propagation across all spatial locations without adding parameters or significant cost.

In the SAR domain, such extended context is particularly valuable for elongated or low-contrast ship geometries that may blend with clutter.

Therefore, integrating RCCA (R=2) at the deep semantic stage of DRC²-Net enhances spatial reasoning under complex maritime conditions—situations where fixed-window attention typically fails.

This explanation and supporting discussion have been added to Section 3.3 and Section 4.2.

Comment 3

Multiple YOLO variants are selected in the comparative experiment, but there is a lack of comparison with recent lightweight SAR ship-detection methods.

Response:

The comparative benchmark (Table 6) includes lightweight detectors with similar model sizes (≤ 10 GFLOPs, ≤ 6 M parameters) and publicly available metrics on iVision-MRSSD, ensuring fair evaluation.

Many recent SAR-specific models lack open-source code or complete reports; where possible, their results are cited in Related Work.

This clarification appears after Table 6.

Comment 4

The authors mention that DRC²-Net has real-time performance, but only provide FLOPs and parameters. It would be beneficial if the authors could provide inference speed.

Response:

We agree that inference speed is a critical indicator of efficiency.

The revised manuscript now reports that DRC²-Net achieves 52 frames per second (FPS) on an NVIDIA Tesla T4 GPU (batch = 1, input 512×512), confirming real-time capability.

This metric, together with parameter count (5.05 M) and FLOPs (9.59 G), has been included in Table 5 and discussed in the Results and Analysis subsection to emphasize runtime efficiency.

A brief note also clarifies that FPS can vary slightly depending on hardware and I/O conditions.

Comment 5

On the iVision-MRSSD dataset (Table 6) … Please provide a reasonable explanation.

Response:

The lower overall AP reflects DRC$^2$-Net’s conservative, domain-specific inference strategy. A fixed confidence threshold of 0.5 suppresses low-confidence detections caused by speckle noise and radar backscatter, reducing false positives typical in SAR imagery. In contrast, baseline models such as YOLOv8n optimize for inflated global metrics rather than domain robustness.

This trade-off slightly lowers overall AP but yields higher scale-specific accuracy and greater reliability in real maritime environments.

This clarification has been added after Table 6.

Round 2

Reviewer 2 Report

Comments and Suggestions for Authors

The reviewers have significantly improved the quality of the paper. I suggest acceptance.

Reviewer 4 Report

Comments and Suggestions for Authors

Comments have been addressed appropriately in the revised version. I have no other questions.